# Growth and development of succulent mixtures for extensive green roofs in a Mediterranean climate

Giuseppe Di Miceli[1], Nicolò Iacuzzi[1], Mario Licata[1]\*, Salvatore La Bella[1], Teresa Tuttolomondo[1], Simona Aprile[2]

1 Department of Agricultural, Food and Forest Sciences, Università degli Studi di Palermo, Palermo, Italy,
2 CREA Research Centre for Plant Protection and Certification, Bagheria (PA), Italy

\* mario.licata@unipa.it

**Data Availability Statement:** All relevant data are within the paper and its Supporting Information files.

## Abstract

Green roof systems, aimed at reducing anthropic impact on the environment, are considered environmental mitigation technologies and adopted by many countries across the world to strengthen urban ecosystem services. This study evaluates two mixtures of succulent: one of *Crassulaceae* and the other of *Aizoaceae*, used in the creation of a continuous and homogenous plant groundcover in Mediterranean environments. To assess the species mixtures, the parameters plant height, growth index, cover percentage and flowering were observed. Hydrological observations were also carried out to evaluate the rainfall retained by the test system in any given month. All data were subjected to analysis of variance. Growth indicators in the study showed trends characteristic of xeric plants, which tend to slow down in dry, summer climate conditions to the point of halting plant vertical growth and ground cover development completely. The *Aizocaeae* mix, during the initial stage, showed prevalent horizontal growth, confirmed by greater a greater growth index (13,21) and cover percentage (45%) compared to *Sedum* (Growth index: 3,61; Cover: 36%). In contrast, the *Sedum* mix recorded greater vertical growth at the beginning (*Sedum* mixture: 7.53 cm; *Aizoaceae* mixture: 6,11 cm). During the final stages of observations, however, greater vertical growth in the *Aizoaceae* (7,88 cm) became apparent together with a recovery in horizontal growth in the *Sedum* (79%), albeit not sufficient to outperform the *Aizoaceae* mixture (87%). Flowering in the two mixtures occurred between late spring and late summer. The *Sedum* mixture guaranteed a longer flowering period (130 days) compared to the *Aizoaceae* (93 days), with a gradual start followed by steady flower emission. Regarding rainfall water retention, a comparison of the mixtures in late winter/early spring revealed that the *Sedum* performed best (44.9 L m$^2$ *vs* 37.4 L m$^2$), whilst the *Aizoaceae* outperformed the *Sedum* in Autumn (63 L m$^2$ *vs* 55 L m$^2$), in conjunction with favorable growth rates in both species mixtures. Both mixtures demonstrated satisfying results and are considered suited to a Mediterranean environment. Furthermore, based on the different growth rates of the species in the two test mixtures, this study suggests that new combinations of *Sedum* and *Aizoaceae* together might prove more resilient in Mediterranean environments.

**Funding:** Title of project: Ar.Co.Verde Grant number: DM 19741/7643/08. Full name of the funder: Italian Ministry of Agricultural, Food and Forestry Policies for funding the project. The funders had no role in study design, data collection and analysis, decision to publish, or preparation of the manuscript.

**Competing interests:** The authors have declared that no competing interests exist.

## Introduction

Over the last decade, practices and government actions in many countries have been directed towards environmental sustainability, aiming to reduce anthropic impact on the environment [1–6]. This has led to the development of environmental mitigation technologies either to reduce $CO_2$ emissions, heat output, air and water pollution or to improve rainwater management. 'Green roofs' or 'living roofs', comprising roofs which are partially or wholly covered by vegetation [7–9], can be considered one such technology. The ability of a green roof to reduce the effects of the urban heat island (with energy saving benefits), to improve the quality of the air, to manage stormwater runoff and to help increase urban biodiversity is widely recognized [10–21]. The effectiveness of a green roof on the environment strongly depends on the local climate, the design of the green roof and the characteristics of the building. In the Mediterranean, in particular, a great deal of attention has been given, over the last ten years or so, to research and development in green roof systems in urban areas [8,22–26]. This is mainly linked to high performance levels obtained by these systems in arid climates, benefitting greatly from the transfer of heat through latent heat processes [27].

The Mediterranean climate is typically characterized by hot, dry summers and cool, humid winters. Annual rainfall generally ranges from 300 to 900 mm/year, with rainfall concentrated mostly in the winter months [28]. Winter temperatures in the Mediterranean Area are generally mild (7–13˚C) with rare occurrences of frost, whilst the summers are typically hot, with average temperatures between 14–25˚C [29]. In general, regions with a Mediterranean climate are positioned along a climate gradient stretching between temperate regions and desert climate regions [30], with conditions which can vary (within specific areas) between mesic and xeric.

Plant species used in the creation of green roof systems need to be particularly adapted, as they play a crucial role in determining the efficiency of the whole system; more so than other components, such as the substrate, particularly in terms of water retention and use [31]. In this regard, further investigation into how green roofs perform seasonally over the long term in a Mediterranean climate [32] is of extreme interest, also as they are considered a tool for rainwater management in urbanized areas [33–36].

In regard to plants which adapt to the abovementioned environmental conditions, succulents are known to perform well in extensive green roof systems in hot-arid climates, both in summer and winter [37]. These plants tolerate cold and drought, including the extreme conditions (high temperatures, high wind, aridity) experienced on a wall or roof [38]; they are efficient plants [39], they have a strong survival capacity and are highly adapted to agamic propagation [40–42]. Finally, they are also considered a good contribution to expanding urban biodiversity [43].

Succulents include over 12500 species, as reported by Nyffeler and Eggli [44], and the *Crassulaceae* and *Aizoaceae* families, with 690 species, are amongst the most widespread and cosmopolitan. These plants are cultivated and distributed worldwide in all types of habitats [45]. They are used not only for ornamental purposes but also due to their beneficial effects on the environmental and health [46].

In various parts of the world, *Sedum* have been used successfully in green roofs to create a thin plant layer which is homogenous and drought resistant [47–49]. *Aizoaceae*, with various species of the genera *Carpobrothus* and *Aptenia*, have been found to provide good groundcover with satisfactory resistance to hot and dry climate conditions in Australia [50,51], and, in tests conducted in Tel Aviv, they provided the best results for cooling [52].

Based on knowledge gained, this research sought to assess two mixtures of succulents, one mixture of *Crassulaceae* and the other of *Aizoaceae*, to create two continuous and homogenous

plant ground cover systems suited to the Mediterranean environment. More specifically, two 'fine-scale' multi-species mixtures were used: four species of the genus *Sedum* were compared with three species of *Aizoaceae*. In these designs, individual plants are typically surrounded by neighbors from different species [53].

The following parameters were observed in order to assess the mixtures: plant height, growth index, cover and flowering. Furthermore, taking into consideration the months in which rainfall events occurred, hydrological observations were also carried out to determine the amount of water retained by the systems in a month.

## Materials and methods

### Test site

Tests were carried out from 2011 to 2012 at the CREA-DC Research farm in Bagheria (Sicily, Italy) (38˚05'00" N– 13˚30'00" E, 78 m a.s.l.). This town is located on the north coast of Sicily in the province of Palermo. The climate in Bagheria is characterized by mild winters and warm, dry summers, and can be considered as representative of Mediterranean coastal areas: the primary focus of this study. (Winter begins in December and ends in March. The winter months are: December, January, February and March. The average maximum temperature of the winter period is 14.9˚C, while the average minimum temperature is 9. 4˚C. The average rainfall of the winter season is 234 mm. Summer starts here at the end of June and ends in September. The summer months are: June, July, August, September. Average maximum summer temperatures are 28,4˚C, the average minimum temperature is 20.9˚C while the average rainfall in the summer season is: 26 mm)

### Description of the test roof system

Tests were carried out in the open air on several pilot roofs using 6 zinc-coated (galvanized) iron platforms, designed and built specifically for the study. The platforms were insulated with 3-cm extruded polystyrene panels. Each platform was 2.2 m$^2$ and a height of 100 cm from the ground in size and supported a lightweight extensive green roof system consisting of:

1. A water drainage layer: a horizontal and vertical ECODREN SD5 layer was used, consisting of a 5.0 mm-thick geonet heat-bonded nonwoven geotextile with a filtering action. This directed water towards a drainage hole positioned at the base of the platform. Each platform was equipped with a leachate collection and measurement system.

2. A water accumulation layer: 5.0 cm-thick calendared non-woven geotextile bags were used containing expanded perlite (AGRILIT®) with grain size of 0.1–1.0 mm.

3. A growth medium for light, large-scale, intensive cover (AgriTERRAM® TVS): this consisted of a mix of peat, lapillus, pumice, zeolites and slow-release fertilizers, weed-seed free with a grain size of 0.0–10 mm and a flat bag thickness of 5.0 cm.

4. Plant-layer types: these consisted of two different types of succulent plant mixtures included in the two study treatments.

### Treatment

The tests compared two succulent plant mixtures sourced from mother plants obtained by the agamic propagation of wild plants (S12 Table). The two mixtures were named *Sedum* mixture and *Aizoaceae* mixture. *Sedum* mixture was a mixture of four *Sedum* species (*S. sediforme*, *S. ochrolecum*, *S. album*, *S. hispanicum*) and Aizoaceae mixture was a mixture of three species

belonging to the *Aizoaceae* Family (*Drosanthemum floribundum*, *Aptenia cordifolia*, *Carpobrotus edulis* (Table 1). The plant m$^{-2}$ investment for each treatment was determined by plant species size and habitus.

## Cultivation practices

Immediately following transplanting, all treatment plots were irrigated with 20 L m$^{-2}$ of tap water using a scale-marked measuring jug. Over the two years of observations, supplementary irrigation events were applied during the dry season (from May to September), providing a total of approx. 80 L m$^{-2}$ of water/year for each plot.

## Plant performances

To compare the two mixtures, a randomized plot design was adopted with 3 replications. The observation sample for the measured parameters of each treatment consisted of the total number of plants. Plants performance was assessed by various indicators (plant height, growth index, cover and flowering) used during the first year (2011). The coverage percentage was recorded starting from 2011 and ended in 2012. During the rainy months of 2012, monthly hydrological observations were also made to assess the capacity of the various mixtures to retain rainfall.

**Plant height.** Plant height was used as an index of vertical growth and measurements were made on a monthly basis during the first year of testing (Feb-Nov 2011). Height, expressed in cm, was defined as the distance from the bottom of the plant to the highest leaf apex [55].

**Growth index.** Growth Index (G.I.) was determined during the first stage (Feb-Nov 2011) to evaluate plant growth. Initial plant growth rates were calculated measuring the height and width of each plant in both directions every 30 days for 10 months. The G.I of each plant was then calculated, as reported by Monterusso et al. [56] and Schaefer et al. [57], by taking the average of 3 measurements, using the following equation: $\frac{H+W1+W2}{2}$ where H is the plant height, W1 is the transversal diameter of the plant, W2 is the longitudinal diameter of the plant.

**Cover.** Cover was used as an index of horizontal plant growth and expressed as a percentage. Plant ground cover was measured twice a month starting from month 6 after transplanting and finishing at month 22 (years 2011–2012), to determine 'mixed plant' ground cover. For this calculation, all plots were photographed with a digital camera located at a distance of

**Table 1. Comparison of the plant species used in the two mixtures.**

| *Sedum* mixture *Crassulaceae* | | | *Aizoaceae* mixture *Aizoaceae* | | |
|---|---|---|---|---|---|
| **Species** | **Plant-life form** | **% in mixture** | **Species** | **Plant-life form** | **% in mixture** |
| *Sedum sediforme* | Ch succ | 24% | *Drosanthemum floribundum* | Ch succ | 60% |
| *Sedum ochroleucum* | Ch succ | 20% | *Aptenia cordifolia* | Ch suffr | 20% |
| *Sedum album* | Ch succ | 28% | *Carpobrotus edulis* | Ch suffr | 20% |
| *Sedum hispanicum* | T scap | 28% | | | |
| **Plants m$^{-2}$**: 23 | | | **Plants m$^{-2}$**: 12 | | |

The chamaephites are woody perennials at the base, with wintering buds placed at a height of the ground between 2 and 30 cm. Therophytes are annual herbaceous plants and survive the adverse season in seed form. Ch succ = Camephyte succulent (plants with specialized stems and / or leaves for storing water); T scap = Terophyte scapose (plants with an erect flower axis and often without leaves); Ch suffr = Camephyte suffruticose (plants in which the herbaceous portions dry annually and only the woody parts remain alive). From: Ellenberg and Muller [54].

150 cm from the cultivated plain. Shutter speeds were set to 'twilight' to avoid shadow and unify contrast. Flash photography was not used. The area of plant ground cover in each plot was calculated by digital image processing. Adobe Photoshop 5.0 version was used to convert the images into grey scale where black was the plant ground cover area and white was the substrate. Cover percentage were calculated using ImageJ software version 1.38, which provides plant ground cover percentages based on the pixels identified in the photographs [58]. 22 months after planting, cover percentage were evaluated as a function of the factors considered.

**Flowering.** Plants were monitored at least three times each week and the dates of "first bloom" and "full bloom" recorded. "First bloom" is defined as the date on which the first flower bud on the plant opens revealing pistils and/or stamens, and "full bloom" as the date on which 95% of the flower buds have opened (i.e., one bud out of twenty has yet to open) [59]. For each treatment, a bloom calendar was completed and the percentage duration of each bloom determined.

**Hydrological observations.** During the second year, to coincide with the rainy months, the volume of water retained by the two succulent plant mixtures was calculated. This was then compared to monthly rainfall volumes to acquire useful data on the water retaining capacity of the two systems.

Rainfall water from the systems was drained off and collected on a monthly basis in scale-marked containers located under the structure. This quantity of water was then subtracted from the known monthly rainfall levels.

## Climatic data

During the test period, rainfall levels and temperatures were recorded using a Stevenson screen located at the CREA Research Center in Bagheria. The Stevenson screen was a white wooden box with a double-louvered design, located a 1.60 m a.s.l. The screen contained thermometers (ordinary, maximum/minimum), a hygrometer, a psychrometer, a dew cell, a barometer, and a thermograph. In this study, this equipment provided data on average daily air temperatures (˚C) and total daily rainfall (mm).

## Statistical analyses

All data were subjected to analysis of variance using the statistical software "Past" (Hammer & Harper–Oslo, Norway) V. 3.16 for Windows. Data on plant height, growth index, cover percentage and total retained water for *Sedum* and for *Aizoaceae* mixture regarding the whole test period were subjected to a repeated measures analysis of the variance (ANOVA). However, data relating to single dates of observations were subjected to a one-way analysis of variance (ANOVA). Both analyses were followed by the Tukey test ($p < 0.05$). Before performing analysis of variance, all percentages were analysed using arcsine transformation. A linear regression analysis was also performed between height and growth index.

# Results

## Climatic data

Over the test period, temperature and rainfall averages were consistent with averages for the Mediterranean climate already defined by several authors [28,29]. Fig 1 shows that, both in 2011 and 2012, the maximum average air temperature was recorded in August.

In 2011, the maximum average air temperature was recorded during the first 10-day period of August (32.2˚C) and in the 2012, during the third 10-day period of the same month (34.5˚C).

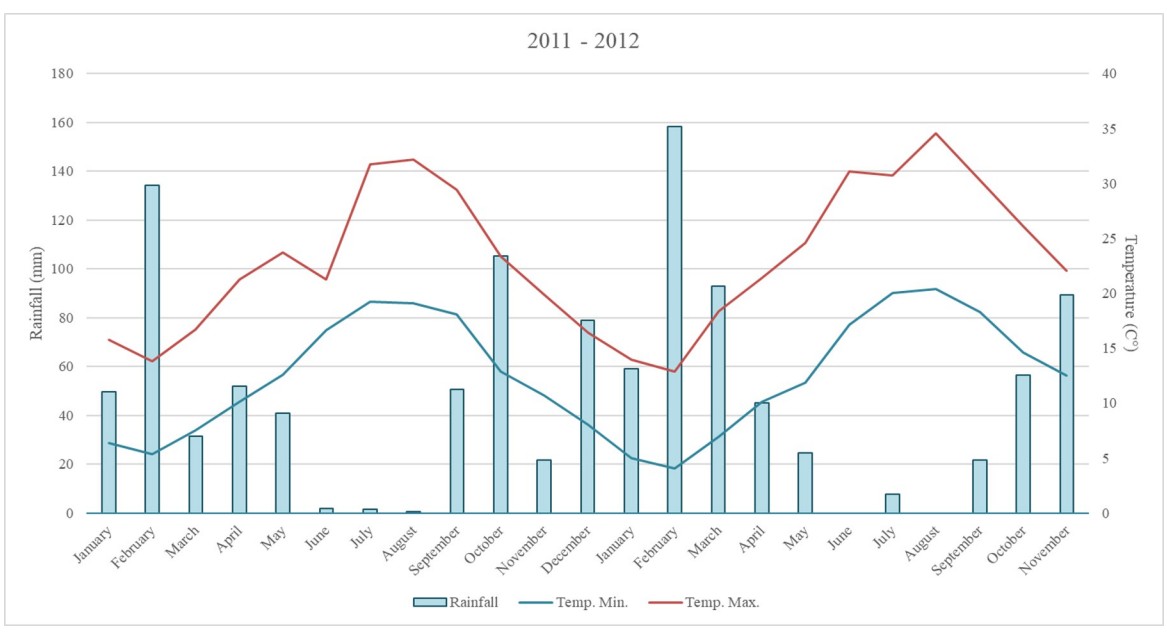

**Fig 1. Rainfall and temperature trends over the test period 2011–2012.**

During both years, minimum average temperatures (below 8˚C) were recorded in the months of January, February and March. Air temperature increased from the beginning of April to the month of August and decreased up to the end of March. The average number of daylight hours recorded during the test period varied over the months, with a minimum of 3.5 to 4.5 hours in December-January and maximum of 10 to 11 hours in June-July. Total rainfall was 568.6 mm in 2011 and 556.8 in 2012. Rainfall was concentrated in the months of February (134 mm) and October (105 mm) in 2011, and in February-March (248 mm) and November (92 mm) in 2012. In both years, the summer period (June-August) was the driest. Average rainfall for the three summer months was 1.26 mm in 2011 and 2.8 mm in 2012.

## Plant height

Fig 2 shows plant height relative to the two succulent mixtures over the first growing season (2011). For this parameter, analysis of the variance (repeated measures ANOVA) for the period February-November 2011, did not reveal any statistically significant differences between the two treatments in the test, with average plant height ranging from 5.70 cm in *Sedum* mixture to 6.03 cm in *Aizoaceae* mixture (Fig 2). As regards one-way ANOVA, however, significant differences were found between the two treatments for some observation dates in the test. More specifically, differences were found for one month after planting (February), with an average plant height for *Aizoaceae* mixture (5.21 cm) which was greater than *Sedum* mixture (3.48 cm); for five months after planting (June), however, this time with *Sedum* mixture (7.53 cm) found to be greater than *Aizoaceae* mixture (6.11 cm); and for the months of September (*Sedum* mixture: 4.96 cm; *Aizoaceae* mixture: 6.79 cm), October (Sedum mixture: 5.91 cm; *Aizoaceae* mixture: 7.31 cm) and November (*Sedum* mixture: 6.71 cm; *Aizoaceae* mixture: 7.88 cm), where significantly higher plants were found in *Aizoaceae* mixture. No significant variations were found between observations in the first and fifth month after planting and between observations in the fifth and eighth month after planting.

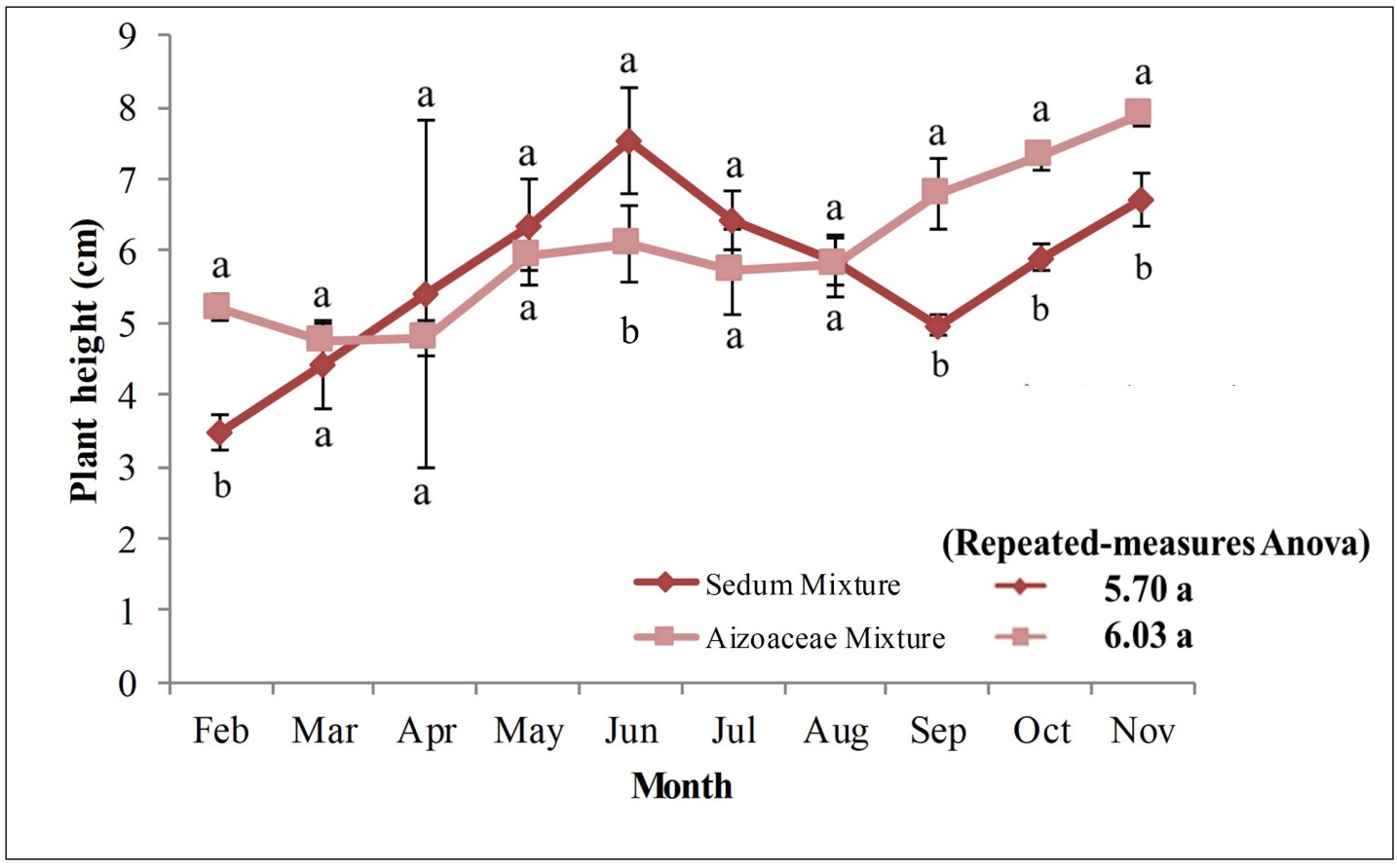

**Fig 2. Plant heights for the two succulent mixtures (Feb-Nov 2011).** Values are means ± SE. For each data, histograms with different letters are significantly different at p≤ 0.05.

## Growth index

Fig 3 shows growth index for the two succulent mixtures during the first growing season (February-November 2011). With average growth index for the period found to be 11.80 in *Sedum* mixture and 20.10 in *Aizoaceae* mixture, significant differences were found between the 2 mixtures (repeated measures ANOVA). Regarding single observations, growth index was consistently significantly higher in *Aizoaceae* mixture for all dates in the test compared to those in *Sedum* mixture.

The *Aizoaceae* mixture, just one month after planting, recorded growth index of 13.21, rising to 26.40 eleven months after planting. In contrast, growth index of 3.81 and 15.90 were found for the two observations (February and November, respectively) for the *Sedum* mixture.

From the second observation date (*Sedum* mixture: 10.31; *Aizoaceae* mixture: 17.81), for both treatments, Growth Index increased up to June (*Sedum* mixture: 13.11; *Aizoaceae* mixture: 20.21), but, at times, with extremely modest growth. From July (*Sedum* mixture: 12.40; *Aizoaceae* mixture: 20.10) to September (*Sedum* mixture: 12.31; *Aizoaceae* mixture: 20.31), growth was negligible, even registering a slight decrease (in *Sedum* mixture in particular) compared to June. In October, however, the growth index increased in both test mixtures (*Sedum* mixture: 15.21; *Aizoaceae* mixture: 23.61), reaching 15.90 in *Sedum* mixture and 26.40 in *Aizoaceae* mixture in November.

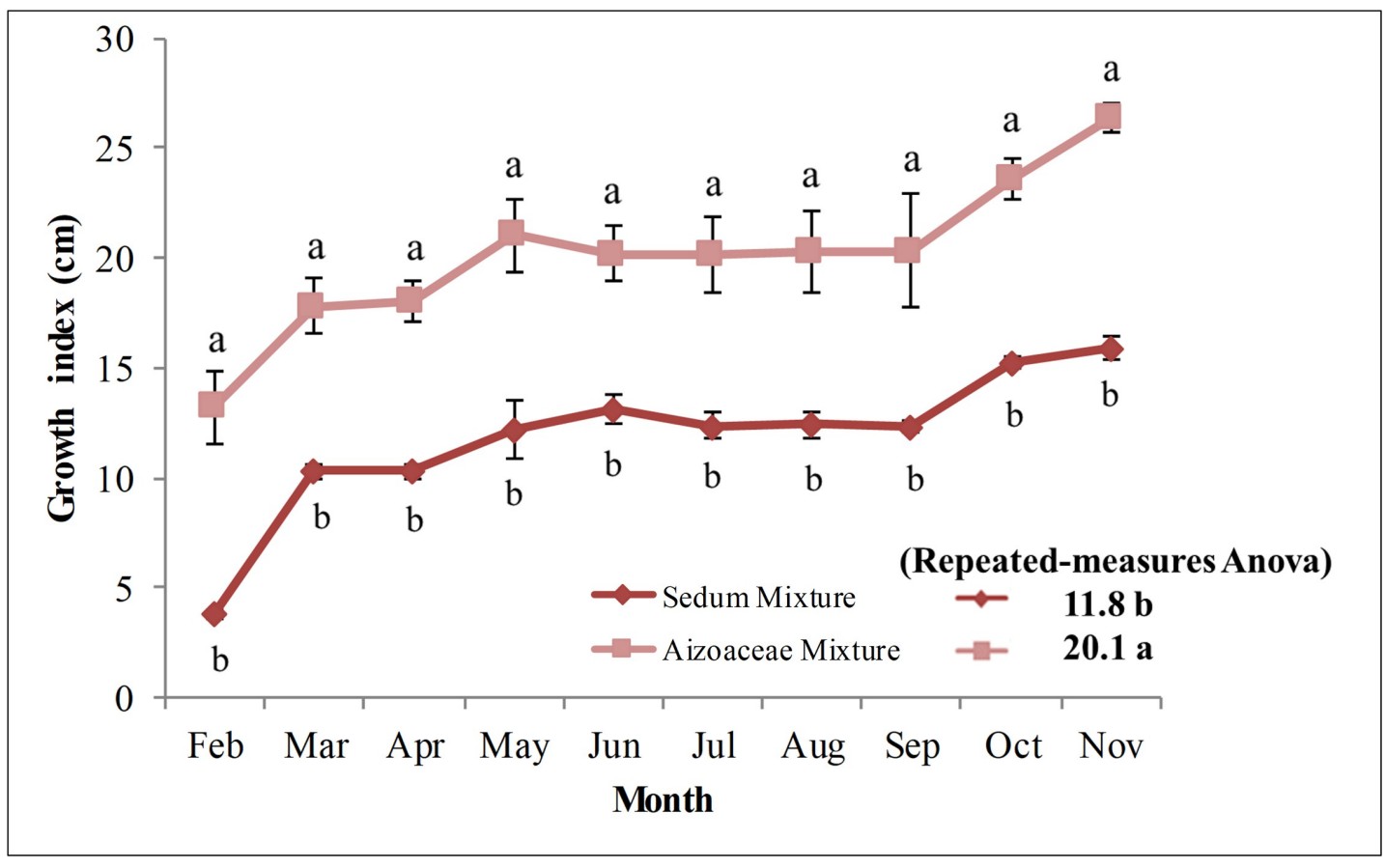

**Fig 3. Plant Growth Index for two succulent mixtures (Feb-Nov 2011).** Values are means ± SE. For each data, histograms with different letters are significantly different at p≤ 0.05.

## Cover

Cover percentage (Fig 4), recorded from month-6 after transplanting (June 2011) up to the end of the second growth season (October 2012), showed significant variations between the two succulent mixtures, with averages of *Sedum* mixture 54.5% and *Aizoaceae* mixture 66.4% (repeated measures ANOVA).

The *Aizoaceae* (*Aizoaceae* mixture) mixtures also produced significantly higher and increasing values for all the observation dates in the study (one-way ANOVA) compared to the *Sedum* (*Sedum* mixture). Regarding the first observation date, *Aizoaceae* mixture had reached a ground cover rate of 45% whilst *Sedum* mixture lagged at approx. 36%, reaching 87% (*Aizoaceae* mixture) and 79% (*Sedum* mixture) by the last observation date.

## Relationship between growth index and plant height

For a more detailed analysis of the relationship between growth index and plant height, linear regression analysis was carried out (Figs 5 and 6). In the two mixtures, growth index increased significantly as plant height increased (*Sedum* mixture: $R^2 = 0.61$, p = 0.007; *Aizoaceae* mixture: $R^2 = 0.70$, p = 0.003). Furthermore, most sensitivity in fluctuations of the two parameters was recorded for mixture *Aizoaceae* mixture, as is clear from the regression line equations (Figs 5 and 6).

**Fig 4. Cover percentage from 6 months to 22 months after transplanting.** Values are means ± SE. For each data, histograms with different letters are significantly different at p≤ 0.05.

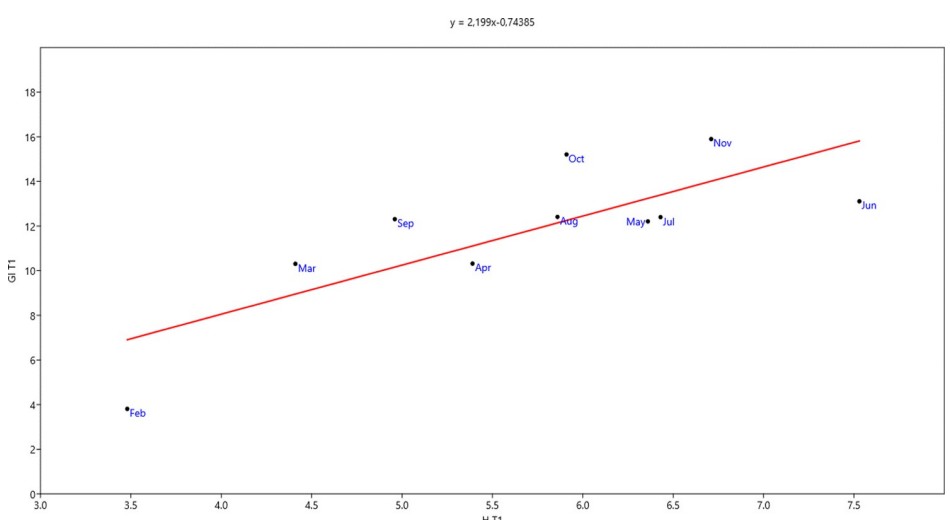

**Fig 5. Relationship between growth index and plant height in *Sedum* mixture.**

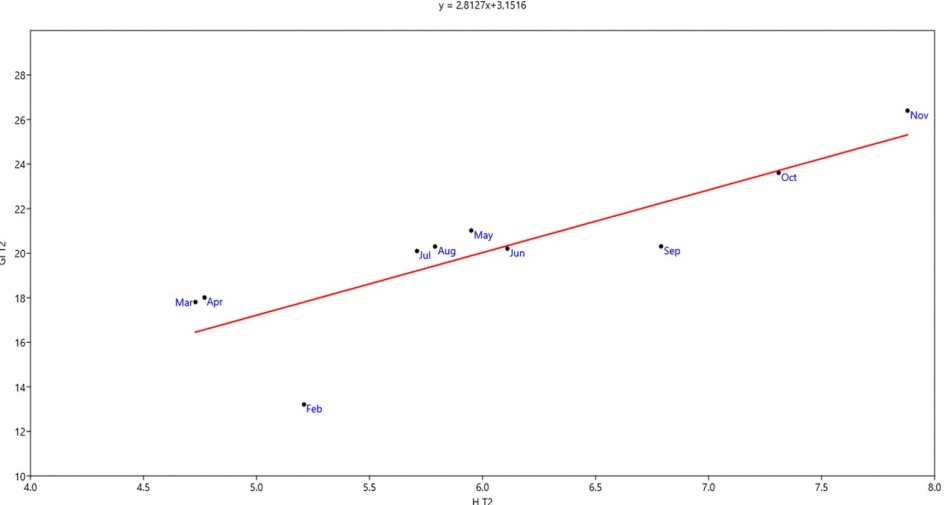

**Fig 6. Relationship between growth index and plant height in *Aizoaceae* mixture.**

## Flowering

Flowering (Fig 7), observed over the course of the first year (2011), began in May for both treatments, with *Aizoaceae* mixture flowering approx. 15 days earlier than *Sedum* mixture. End of flowering was established as mid-September for *Sedum* mixture, whilst *Aizoaceae* mixture had already stopped flowering at the end of July. Observations also showed that flowering onset in *Sedum* mixture was more gradual, with values 15–30% of open flowers recorded between May and June, reaching full bloom towards the beginning of July with 95% of open flowers (Fig 8). In contrast, *Aizoaceae* mixture immediately provided a much more abundant bloom, reaching full bloom between May and June with 80–100% of open flowers; however, flowering was over by the end of July.

## Hydrological observations

Fig 9 shows the amount of monthly rainfall retained by the two systems during the second year of growth (January–November 2012). Analysis of the averages for all of the test periods did not provide any significant differences between the two mixtures (retained water *Sedum*: 24.9 L m$^{-2}$ andretained water *Aizoaceae*: 23.6 L m$^{-2}$; repeated measures ANOVA). Single observations, however, revealed significant differences for February (retained water *Sedum*:

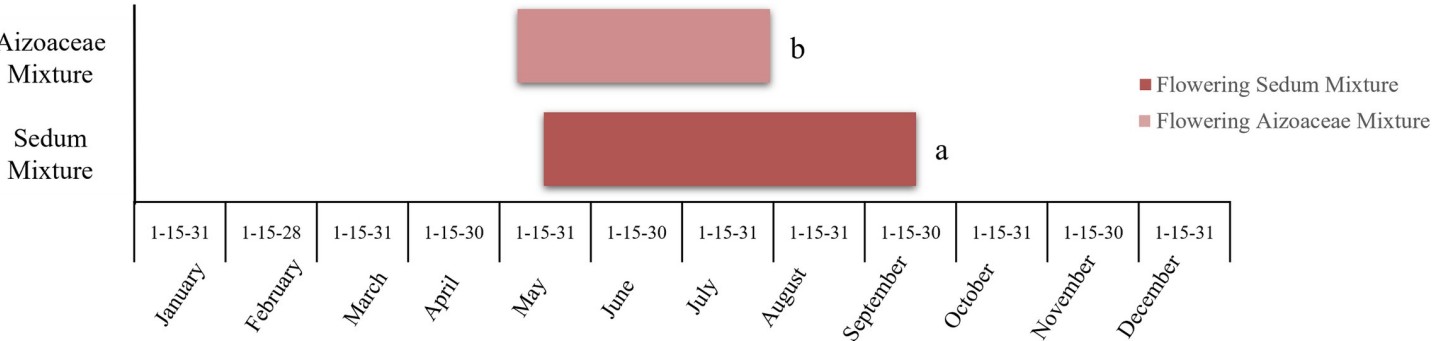

**Fig 7. Duration of flowering stage in the two test treatments–Year 2011.**

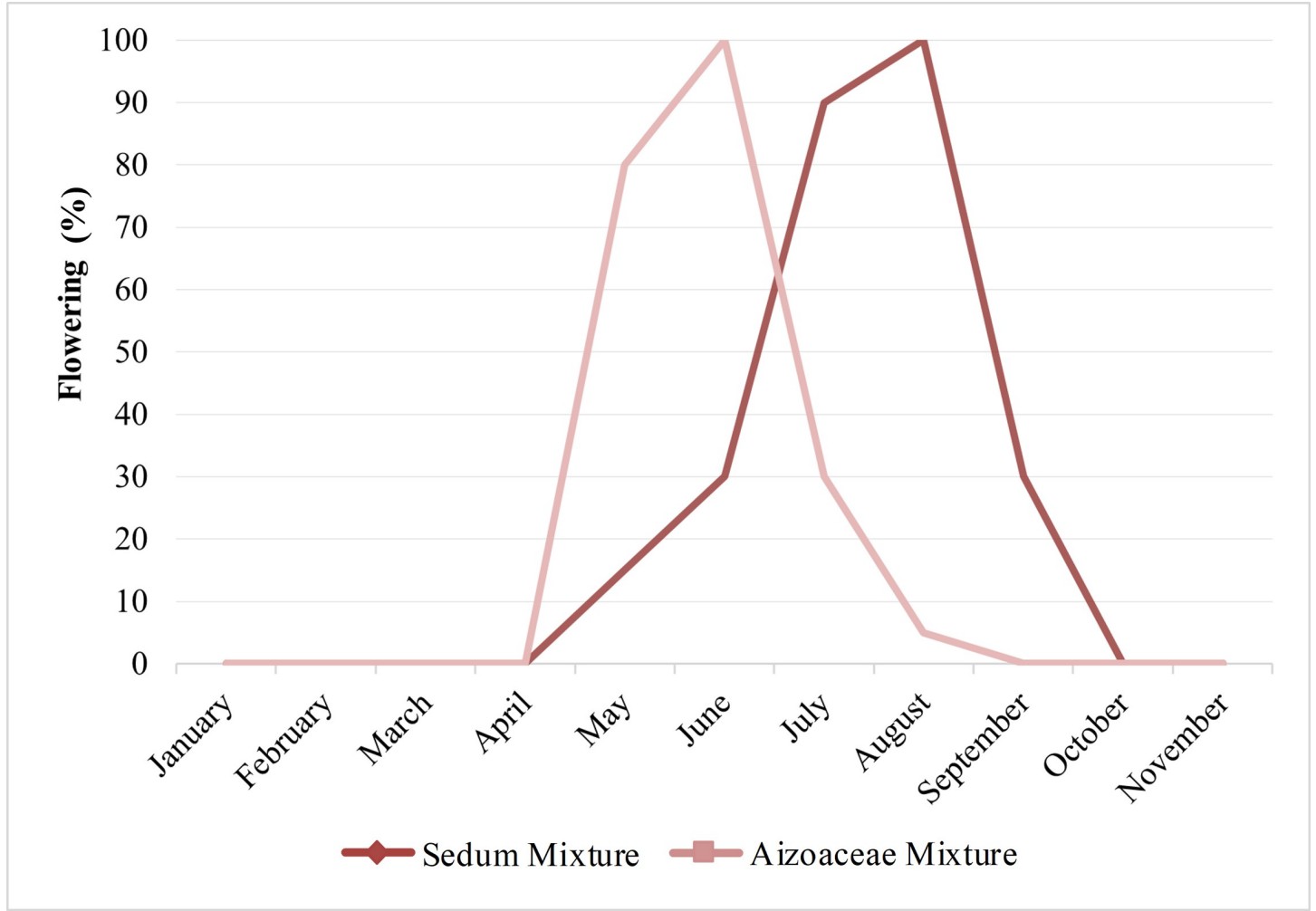

**Fig 8. Bloom percentage in the two test treatments–Year 2011.**

44.9 L m$^{-2}$ vs retained water *Aizoaceae*: 37.4 L m$^{-2}$) and March (retained water *Sedum*: 29 L m$^{-2}$ vs retained water *Aizoaceae*: 17 L m$^{-2}$) in favour of *Sedum* mixture, and November (retained water *Sedum*: 55 L m$^{-2}$ vs retained water *Aizoaceae*: 63 L m$^{-2}$) in favour of *Aizoaceae* mixture. No significant differences were found for the remaining months of observations.

## Discussion

Results obtained during activities show that succulents are, in general, suited to use in green roofs in the Mediterranean area, managing to grow in the given test conditions with low maintenance input, as also found by other authors in similar conditions [38,56]. Typical of hot-arid and desert climates, the *Aizoaceae* and *Crassulaceae* used in the test may have found environmental conditions which are not dissimilar to their original environment [48].

In general, analysis of the growth indicators shows growth rates which are typical of xerophytes. During dry, summer climate conditions, plant vertical growth and development of ground cover is reduced to a halt.

With the exception of a significant initial lead in plant height recorded in the *Aizoaceae* mixture, the two mixtures then began to show similar growth trends, with plant heights for

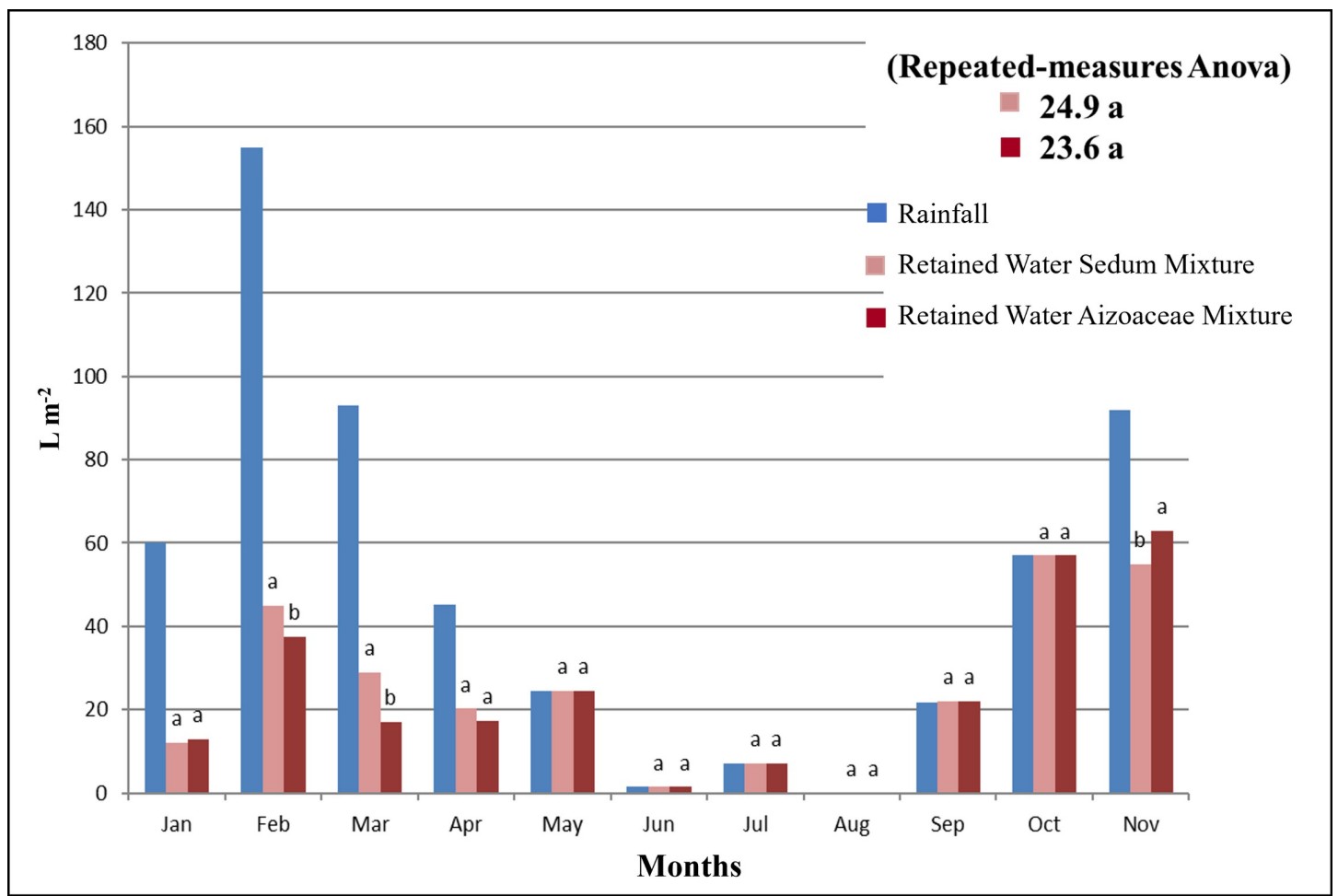

**Fig 9. Rainfall retained by the systems year 2012.** For each data, histograms with different letters are significantly different at p≤ 0.05.

*Aizoaceae* mixture recorded as lower than *Sedum* mixture, but with no statistical differences. It is worth noting that, compared to *Aizoaceae* mixture, *Sedum* mixture showed continual vertical growth right up to June, performing more favourably in this aspect than the other mixture. However, in successive months, both mixtures (*Sedum* mixture and *Aizoaceae* mixture) slowed to a halt, even witnessing a reduction in average plant height compared to preceding months. This reduction in size, which may seem a little unexpected for this parameter, is due to the apexes drying out in the high temperatures and to the lack of water, typical of Mediterranean environments in this period.

This reduction in size was seen to be greater in the *Sedum* up to September, whilst the *Aizoaceae* mixture recovered vertical growth in August, with development which was statically greater than *Sedum* mixture from September to November.

The *Aizoaceae* mix proved constantly higher for growth index and cover percentage than the *Sedum* mix regarding both every sample date and in relation to the whole growth season. The *Aizoaceae* mix had already reached 70% of cover just 15 months after transplanting, extending to 80% in approx. 20 months.

The *Sedum* mix, although not reaching 80% cover in the 20 months of observation, did obtain a similar profile to the other mixture. Although slightly less developed, it was, however,

considered suited to the test conditions, as it managed to ensure a certain degree of cover percentage, above all in the initial stages due to greater vertical growth than the *Aizoaceae*. The *Sedum* mixture obtained 80% cover just two months later (at 22 months), when the *Aizoaceae* mixture was reaching 90% cover. It is worth pointing out that the presence of *Sedum hispanicum* in the *Sedum* mixture, due to its annual behaviour, may not have contributed well to expected levels of plant cover. Schindler et al. [60] reported that when the aim is to obtain as extensive a cover as possible, the use of perennial *Sedum* species may be more suited, rather than annuals. However, it is also true that annuals are able to contribute to the floral diversification of the systems, not only in terms of biodiversity in general but also within the same mix [42].

An explanation for the more favorable growth index and percentage cover results obtained by the *Aizoaceae* in comparison to the *Sedum* mix may be given by the morpho-physiological characteristics of the species selection. *Carpobrothus edulis*, with its long, thick stalks, may have given a significant contribution to horizontal growth cover, and, similar to data reported by Razzaghmanesh [51] regarding *Carpobrothus rossii*, it may have coped better with the hot, dry summers than other species thanks to a more efficient use of water. Similarly, *Aptenia cordifolia* is able to survive long periods of severe environmental conditions which would inhibit growth in other plants [61,62], by adopting various resistance strategies to short-term water shortages, amongst which the ability to control photosynthesis [52].

Contributing to the creation of greater plant cover by the *Aizoaceae* was the presence of *Drosanthemum floribundum* in the mixture, as it forms a dense groundcover. The prevalent horizontal growth of the *Aizoaceae*, above all in the initial stage, is confirmed by superior growth index and percentage cover results compared to *Sedum*, which showed better initial vertical growth. In the end stage of observations, vertical growth was greater in the *Aizoaceae*; horizontal growth in the *Sedum* also recovered, although not enough to outperform the *Aizoaceae* mixture. In this regard, parameters of the linear regression lines between growth index and plant/height allowed us to estimate, for the given test period, an increase in growth index of approx. 2 cm for *Sedum* mixture and approx. 3 cm for *Aizoaceae* mixture for each unit increase of plant/height.

In addition to growth rates, flowering of the two mixtures (concentrated between late spring and the summer) also highlighted the Mediterranean nature of the species in the study. In particular, the *Sedum* mixture ensured a longer flowering period, with a gradual beginning and constancy in the production of flowers throughout the period. This is an interesting feature, according to Nagase and Tashiro-Ishii [63], when deciding upon the selection of species for green roofs which are oriented more towards criteria such as landscape aesthetics or the preservation of rare species, thereby promoting greater plant biodiversity in green roof systems.

Observations on the rainfall capture and retention capacity of the two systems, during the second season, allowed us to make a primary assessment of the water retaining capacity of succulent green roofs in the Mediterranean, also in terms of survival prospects, plant development and groundcover.

Below 60 mm of rain per month, as can be seen in graph 9, the two systems behaved in a similar way, not demonstrating significant differences between the two, although greater rain capture was observed at times in the *Sedum* mixture. The systems differed, however, for rainfall over 80 mm; in February and March mixture *Sedum* mixture produced the most promising results (retained water *Sedum*: 45%; retained water *Aizoaceae*: 38%), and in November, mixture *Aizoaceae* mixture performed best (retained water *Aizoaceae*: 69%; retained water *Sedum*: 60%). From May, with only 25 mm of rainfall/month, no outflow was observed due to the fact that 100% of the rainfall was retained in the system, as with the other summer months.

Retained water from the two succulent systems was not constant during the year, as found also by other authors [64]. Variations were dependent upon the main climate parameter trends, plant growth (vertical and horizontal), activity levels and subsequent shape (one or two dimensional). In this regard, with a comparable structure and climate, the best performance results for the parameter retained water were observed at the end of winter/beginning of spring in the *Sedum* mixture, and in Autumn, in the *Aizoaceae* mix (in conjunction with favorable growth rates in each mixture, as previously described).

The fact that green roofs are dynamic as regards biomass, plant height and cover, moving over time, is well documented in scientific literature [16,65,66]. It is also worth noting that two-dimensional plant ground cover in green roofs is important in terms of visual attractiveness and other ecosystems services [67,68]. As reported by some authors, although the simultaneous presence of species and different growth shapes contributes to maximizing the provision of services on a green roof (reduction of substrate surface temperature, rainfall retention etc.), it is not clear how greater diversity in shape behaves in different climates [66,69,70]. Therefore, greater research in this area is fundamental and this study contributes to furthering knowledge on these aspects, together with the possibility of broadening the range of succulent species which can be considered in the creation of extensive green roofs in the Mediterranean to maximize the functioning capacity of the systems.

## Conclusions

The results of this study show firstly that succulents are suited, in general, to use in green roofs in Mediterranean environments, managing to grow in the test conditions with low input maintenance. More specifically, both succulent mixtures performed to satisfying levels and can be deemed as the correct choice for the Mediterranean and a possible, advantageous solution not only to mitigate summer temperatures, and therefore, improve energy consumption in buildings, but also to capture and retain rainfall. Other benefits include increasing urban plant biodiversity and low-maintenance green areas for citizens, together with potential development of the production sector (nurseries, construction of technological systems etc.) linked to green technologies. Furthermore, this study, based on the different growth rates of the species in the two test mixtures, suggests that new mixtures of *Sedum* and *Aizoaceae* together might prove more resilient in Mediterranean environments.

## Supporting information

**S1 Fig. Pilot roof system.**
(PDF)

**S1 Table. Cover Plant_ *Aizoaceae* (%).**
(XLSX)

**S2 Table. Height Plant _*Aizoaceae* (cm).**
(XLSX)

**S3 Table. Height Plant_ *Sedum* (cm).**
(XLSX)

**S4 Table. Cover (%).**
(XLSX)

**S5 Table. Plant height.**
(XLSX)

**S6 Table. Growth index.**
(XLSX)

**S7 Table. Flowering.**
(XLSX)

**S8 Table. Relationship between G.I. and plant height.**
(XLSX)

**S9 Table. Hydrological observations.**
(XLSX)

**S10 Table. Cover plant_ Sedum (%).**
(XLSX)

**S11 Table. Plant material.**
(XLSX)

**S12 Table. Plant location.**
(XLSX)

## Acknowledgments

The authors would like to thank Dr GianVito Zizzo for having contributed to the research and Lucie Branwen Hornsby for her linguistic assistance.

## Author Contributions

**Conceptualization:** Giuseppe Di Miceli, Nicolò Iacuzzi, Mario Licata, Simona Aprile.

**Data curation:** Teresa Tuttolomondo, Simona Aprile.

**Formal analysis:** Nicolò Iacuzzi, Simona Aprile.

**Funding acquisition:** Simona Aprile.

**Investigation:** Salvatore La Bella, Teresa Tuttolomondo, Simona Aprile.

**Methodology:** Giuseppe Di Miceli, Salvatore La Bella, Simona Aprile.

**Project administration:** Simona Aprile.

**Resources:** Giuseppe Di Miceli.

**Software:** Mario Licata, Salvatore La Bella, Teresa Tuttolomondo.

**Supervision:** Salvatore La Bella, Teresa Tuttolomondo.

**Validation:** Teresa Tuttolomondo.

**Visualization:** Nicolò Iacuzzi, Mario Licata, Salvatore La Bella.

**Writing – original draft:** Nicolò Iacuzzi, Mario Licata, Teresa Tuttolomondo, Simona Aprile.

**Writing – review & editing:** Giuseppe Di Miceli.

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
