## [Decision Letter · Decision Letter 0]

11 Apr 2022

PONE-D-22-05525Growth and development of succulent mixtures for extensive green roofs in a Mediterranean climatePLOS ONE

Dear Dr. Licata,

Thank you for submitting your manuscript to PLOS ONE. After careful consideration, we feel that it has merit but does not fully meet PLOS ONE’s publication criteria as it currently stands. Therefore, we invite you to submit a revised version of the manuscript that addresses the points raised during the review process.

We look forward to receiving your revised manuscript.

Kind regards,

Sajid Ali

Academic Editor

PLOS ONE

Journal Requirements:

Reviewers' comments:

Reviewer 1 

Overall, I enjoyed reading this paper. It was nice to see that several species will be usable in the Mediterranean climate. I have several recommendations for improving clarity. Additionally, I am uncertain as to why the height and flowering of the 3 species of sedum and the 3 species of Aizoaceae were averaged together. I would like to see their range as individual species. I would also like to know the final cover of individual species.

Introduction

Line 68: recommend removing “in these climate conditions” as vegetation can influence ecosystem services regardless of climate. or add more detail as to in what way they must be adapted in comparison to other climates.

Line 76: define what you mean by extreme conditions

Line 84: I recommend adding an example supporting the previous sentence

Methods

Test site: please include precipitation and temp averages for the seasons mentioned. Also how long do your summers/winters last?

Line 121: the term “two plant layer types” is a bit confusing here as it sounds like each panel had two plant layers? Recommend just calling it plant layer as you define your plant treatments in the next section.

Plant Material: Since plos one advertises to a wide variety of researchers I recommend removing species authority from this section and just adding it to one of your appendix tables. Additionally for this journal readers may not know the terms Camephyte succulent; Terophyte scapose; Camephyte suffruticose. I recommend defining.

Line 125: please provide information on location of wild plants, also are the same specieas all from the same mother plant?

Line 125: for ease of understanding recommend changing treatment names from T1 and T2 to Sedum mixture and Aizoaceae mixture

Please explain why species were planted at different densities.

I recommend changing the heading plant material to treatments

I recommend removing the heading cultural practices (implies social practices) as this paragraph can fit just fine in the previous section.

Line 141: confusing sentence, not sure what data was gathered

Line 142: please define various indicators

Line 143: how were hydrological observations made?

Plant height: was this taken for every individual in each treatment?

Line 157: Recommend changing “TPGC” to “cover” to make is easier to remember what it is associated with.

Line 176: how was this calculated?

The terms RW1 and RW2 I recommend spelling out to improve readability. So when mentioned later on just say retention sedum or something similar

For readability Recommend spelling out GI and not using acronym.

Results

No need to write out detailed climatic conditions, recommend just saying similar to previous studies, followed expected trends for region. Recommend moving figure 1 to appendix.

Uncertain why relationship between growth rate and height is important to include. Recommend removing or moving to appendix.

Figures

2-4: uncertain what the numbers under the heading repeated measures anova are

Please add a photograph of study system

Reviewer 2

Abstract

Additional information of statistical analysis and results values must be included in abstract in order to enhance the first look of manuscript.

Materials and methods

What was the height of pilot roof from the ground?

How mixtures obtained from the mother plant.

What was age of mother plant

How plants were propagated.

What type of water was used for irrigation e.g. tap or distilled water

Discussion

Discussion is corroborative. The authors should highlight the reason of their result findings in the light of available literature.

The author did not follow the logical trend to reach the purposes of the manuscript.

Figures

Please use similar writing format in the figures

Clearly write figure No. of all figures

---

## [Author Response · Author response to Decision Letter 0]

16 May 2022

In response to Ms. Ref. No.: PONE-D-22-05525 we followed all the recommendations which were made by the editor and reviewers. 

Editor’s comments

Journal Requirements:

The authors confirm that the manuscripts meets PLOS ONE’s style requirement. They have used the PLOS ONE templates, found at links suggested by the Editor.

The authors agree with the Editor’s consideration. They state that title of project is Ar.Co.Verde, the Grant number is DM 19741/7643/08, the full name of the funder is Italian Ministry of Agricultural, Food and Forestry Policies.

The authors agree with the Editor’s comment. They affirm that the minimum dataset has been included in the Support Information File by tables S1, S2, S3, S4, S5, S6, S7, S8, S9 and S10.

Reviewer #1

In response to Ms. Ref. No.: PONE-D-22-05525 Reviewer #1, we tried to follow nearly all of his/her recommendations. Here is a point by point summary of the actions taken in response to the reviewer’s comments.

*The reviewer wrote: Overall, I enjoyed reading this paper. It was nice to see that several species will be usable in the Mediterranean climate. I have several recommendations for improving clarity. Additionally, I am uncertain as to why the height and flowering of the 3 species of sedum and the 3 species of Aizoaceae were averaged together. I would like to see their range as individual species. I would also like to know the final cover of individual species.

The authors appreciate and thank the reviewer for his/her comments. The height and flowering of the 3 species of sedum and the 3 species of Aizoaceae were averaged together because the aim of the work involved a comparison between two mixtures. The assessment of individual species was not of interest to the authors. However, we enclose the available data relating to the individual species in the Supporting Information (Tables S1, S2, S3, S4, S5, S7 and S10).

Introduction

*The reviewer wrote: Line 68: recommend removing “in these climate conditions” as vegetation can influence ecosystem services regardless of climate. or add more detail as to in what way they must be adapted in comparison to other climates.

The authors thank the reviewer for his/her suggestion. They affirm that the statement “in these climate conditions” has been removed from the manuscript.

*The reviewer wrote: Line 76: define what you mean by extreme conditions

The authors have specified and reported in the manuscript what is meant by "extreme conditions" (high temperatures, high wind, aridity).

*The reviewer wrote: Line 84: I recommend adding an example supporting the previous sentence

The authors have included a bibliographic reference to support the sentence, as suggested by the reviewer.

Methods

*The reviewer wrote: Test site: please include precipitation and temp averages for the seasons mentioned. Also how long do your summers/winters last?

The authors have added in the manuscript the average rainfall and temperature trends of the area, as suggested by the reviewer.

*The reviewer wrote: Line 121: the term “two plant layer types” is a bit confusing here as it sounds like each panel had two plant layers? Recommend just calling it plant layer as you define your plant treatments in the next section.

The authors have changed the sentence in the manuscript, as suggested by the reviewer.

*The reviewer wrote: Plant Material: Since plos one advertises to a wide variety of researchers I recommend removing species authority from this section and just adding it to one of your appendix tables. Additionally for this journal readers may not know the terms Camephyte succulent; Terophyte scapose; Camephyte suffruticose. I recommend defining.

The authors agree with the reviewer’s observation. They have made a table and inserted it in the Supporting Information (Table S11) and defined the required terms.

*The reviewer wrote: Line 125: please provide information on location of wild plants, also are the same specieas all from the same mother plant?

The authors thank the reviewer for this constructive comment. The authors have added the concerning the location in the Supporting Information (Table S12). They highlight that each species was from mother plants of the same population.

*The reviewer wrote: Line 125: for ease of understanding recommend changing treatment names from T1 and T2 to Sedum mixture and Aizoaceae mixture

The authors have changed the names in the manuscript, as suggested by the reviewer. 

*The reviewer wrote: Please explain why species were planted at different densities.

This information is provided in line 130, "The plant m-2 investment for each treatment was determined by plant species size and habitus".

*The reviewer wrote: I recommend changing the heading plant material to treatments

The authors have changed the name, as suggested by the reviewer.

*The reviewer wrote: I recommend removing the heading cultural practices (implies social practices) as this paragraph can fit just fine in the previous section.

The authors have changed the name from Cultural practices to “Cultivation practices” in order to explain better the sentence.

*The reviewer wrote: Line 141: confusing sentence, not sure what data was gathered

The sentence has been changed and improved in the manuscript in order to avoid any confusion.

*The reviewer wrote: Line 142: please define various indicators

The various indicators have been defined in the manuscript. "Plant height, Growth index, Cover, Flowering."

*The reviewer wrote: Line 143: how were hydrological observations made?

The hydrological observations have been described on lines 175-180 as following. “During the second year, to coincide with the rainy months, the volume of water retained by the two succulent plant mixtures was calculated. This was then compared to monthly rainfall volumes to acquire useful data on the water retaining capacity of the two systems. Rainfall water from the systems was drained off and collected on a monthly basis in scale-marked containers located under the structure. This quantity of water was then subtracted from the known monthly rainfall levels ".

*The reviewer wrote: Plant height: was this taken for every individual in each treatment?

The height of the plant was measured for every individual in each treatment.

*The reviewer wrote: Line 157: Recommend changing “TPGC” to “cover” to make is easier to remember what it is associated with.

The authors have changed the name, as suggested by the reviewer.

*The reviewer wrote: Line 176: how was this calculated?

The volume of water retained by the two systems was calculated by the difference between the height of rain and the water flowed from the systems which was collected by graduated containers placed at the base of the system.

*The reviewer wrote: The terms RW1 and RW2 I recommend spelling out to improve readability. So when mentioned later on just say retention sedum or something similar.

The authors have changed the sentence, as suggested by the reviewer.

*The reviewer wrote: For readability Recommend spelling out GI and not using acronym.

The acronyms have been replaced with the full name.

Results

*The reviewer wrote: No need to write out detailed climatic conditions, recommend just saying similar to previous studies, followed expected trends for region. Recommend moving figure 1 to appendix.

The authors thank the reviewer for his/her comment. However, they have not changed the paragraph because of the importance that climate conditions assume in the test environment.

*The reviewer wrote: Uncertain why relationship between growth rate and height is important to include. Recommend removing or moving to appendix.

The authors thank the reviewer but would like to leave the paragraph at the end to deepen the relationship between the two parameters.

Figures

*The reviewer wrote: 2-4: uncertain what the numbers under the heading repeated measures anova are

The values under the “heading ANOVA repeated measures” represent the averages of the parameter of the two thesis with reference to the entire period.

*The reviewer wrote: Please add a photograph of study system

The authors have added the a photograph in the Supporting information (Figure S1).

Reviewer #2

In response to Ms. Ref. No.: PONE-D-22-05525 Reviewer #2, we tried to follow nearly all of his/her recommendations. Here is a point by point summary of the actions taken in response to the reviewer’s comments.

Abstract

*The reviewer wrote: must be included in abstract in order to enhance the first look of manuscript.

The authors thank the reviewer for hisher comment and have included the additional information of statistical analysis and results values in the abstract. 

Materials and methods

*The reviewer wrote: What was the height of pilot roof from the ground?

The authors have reported this information in the manuscript: "Each platform was 2.2 m2 and a height of 100 cm from the ground in size and ..."

*The reviewer wrote: How mixtures obtained from the mother plant.

This information has been provided in line 130, "The plant m-2 investment for each treatment was determined by plant species size and habitus".

*The reviewer wrote: What was age of mother plant

The age of the mother plants, obtained from wild plants, was 3 years.

*The reviewer wrote: How plants were propagated.

The plants were obtained by agamic propagation.

*The reviewer wrote: What type of water was used for irrigation e.g. tap or distilled water

Tap water was used for the irrigation.

Discussion

*The reviewer wrote: Discussion is corroborative. The authors should highlight the reason of their result findings in the light of available literature.

The authors thank the reviewer for his/her constructive consideration. They have improved the manuscript with recent references.

*The reviewer wrote: The author did not follow the logical trend to reach the purposes of the manuscript.

The authors think they have discussed the parameters characterizing the green roof following a logical trend.

Figures

*The reviewer wrote: Please use similar writing format in the figures

The format of the figures has been re-written and improved.

*The reviewer wrote: Clearly write figure No. of all figures

The caption of the figures has been clearly re--written.

We hope the editorial board will agree on the interest of this study.

Sincerely yours,

Mario Licata on behalf of the authors.

Corresponding author: Dott. Mario Licata, Department of Agricultural, Food and Forest Sciences, Università degli Studi di Palermo, Viale delle Scienze 13 Building 4, 90128 Palermo, Italy. E-mail: mario.licata@unipa.it

---

## [Editor Report · Decision Letter 1]

23 May 2022

Growth and development of succulent mixtures for extensive green roofs in a Mediterranean climate

PONE-D-22-05525R1

Dear Dr. Licata,

We’re pleased to inform you that your manuscript has been judged scientifically suitable for publication and will be formally accepted for publication once it meets all outstanding technical requirements.

Kind regards,

Sajid Ali

Academic Editor

PLOS ONE
---

## [Editor Report · Acceptance letter]

25 May 2022

PONE-D-22-05525R1 

Growth and development of succulent mixtures for extensive green roofs in a Mediterranean climate 

Dear Dr. Licata:

I'm pleased to inform you that your manuscript has been deemed suitable for publication in PLOS ONE. Congratulations! Your manuscript is now with our production department. 

Kind regards, 

on behalf of

Dr. Sajid Ali 

Academic Editor

PLOS ONE